# Personalizing the decision of dabigatran versus warfarin in atrial fibrillation: A secondary analysis of the Randomized Evaluation of Long-term anticoagulation therapY (RE-LY) trial

**Samuel W. Reinhardt**[1☯]\*, **Nihar R. Desai**[1,2☯], **Yuanyuan Tang**[3], **Philip G. Jones**[3], **Jeremy Ader**[4¤], **John A. Spertus**[3,5]

**1** Section of Cardiovascular Medicine, Department of Internal Medicine, Yale School of Medicine, New Haven, Connecticut, United States of America, **2** Center for Outcomes Research and Evaluation, Yale-New Haven Hospital, New Haven, Connecticut, United States of America, **3** Saint Luke's Mid America Heart Institute, Kansas City, Missouri, United States of America, **4** Yale School of Medicine, New Haven, Connecticut, United States of America, **5** Section of Cardiovascular Disease, Department of Internal Medicine, University of Missouri-Kansas City, Kansas City, Missouri, United States of America

☯ These authors contributed equally to this work.
¤ Current address: Department of Neurology, Columbia University Medical Center, New York, New York, United States of America
\* samuel.reinhardt@yale.edu

**Data Availability Statement:** The data are available through application to Boehringer-Ingelheim, and

## Abstract

### Background

The RE-LY (Randomized Evaluation of Long-Term Anticoagulation Therapy) trial demonstrated that higher-risk patients with atrial fibrillation had lower rates of stroke or systemic embolism and a similar rate of major bleeding, on average, when treated with dabigatran 150mg compared to warfarin. Since population-level averages may not apply to individual patients, estimating the heterogeneity of treatment effect can improve application of RE-LY in clinical practice.

### Methods and results

For 18040 patients randomized in RE-LY, we used patient-level data to develop multivariable models to predict the risk for stroke or systemic embolism and for major bleeding including all three treatment groups (dabigatran 110mg, dabigatran 150mg, and warfarin) over a median follow up of 2.0 years. The mean predicted absolute risk reduction (ARR) for stroke/systemic embolism with dabigatran 150mg compared to warfarin was 1.32% (range 11.6% lower to 3.30% higher risk). The mean predicted ARR for bleeding was 0.41% (range 8.93% lower to 63.4% higher risk). Patients with increased stroke/systemic embolism risk included those with prior stroke/TIA (OR 2.01), diabetics on warfarin (OR 2.00), and older patients on dabigatran 150mg (OR 1.68 for every 10-year increase). Major bleeding risk was higher in patients on aspirin (OR 1.25), with a history of diabetes (OR 1.34) or prior

we as the authors do not have jurisdiction to share the data ourselves. Requests for data access can be made through Vivli (https://vivli.org/). The authors in this study did not have any special access privileges that others would not have.

**Funding:** The author(s) received no specific funding for this work.

**Competing interests:** I have read the journal's policy and the authors of this manuscript have the following competing interests: Nihar Desai works under contract with the Centers for Medicare and Medicaid Services to develop and maintain performance measures used for public reporting and pay for performance programs. He reports consulting for Amgen, Boehringer Ingelheim, Cytokinetics, Relypsa, Novartis, and SCPharmaceuticals. John Spertus reports an equity interest in Health Outcomes Sciences and consulting for Amgen, Bayer, Merck, Myokardia, Novartis, United Healthcare, BCBS of Kansas City and ownership of the copyright to the SAQ, KCCQ and PAQ. All other authors report no disclosures. This does not alter our adherence to PLOS ONE policies on sharing data and materials.

stroke/TIA (OR 1.22), those with heart failure on dabigatran 110mg (OR 1.52), older patients on either dabigatran 110mg or 150mg (OR 1.57 and 1.93, respectively, for each 10-year increase), and heavier patients on dabigatran 110mg or 150mg; patients in a region outside the United States and Canada and with better renal function had lower bleeding risk.

## Conclusions

There is substantial heterogeneity in the benefits and risks of dabigatran relative to warfarin among patients with atrial fibrillation. Using individualized estimates may enable shared decision making and facilitate more appropriate use of dabigatran; as such, it should be prospectively tested.

## Clinical trial registration

www.clinicaltrials.gov number, NCT00262600.

## Introduction

Applying data from clinical trials to everyday practice is a significant challenge in the current age of evidence-based medicine. Large trials can demonstrate significant differences in mean outcomes for a given treatment, but such estimates ignore the heterogeneity of treatment effect among different patients within the study population. Subgroup analyses try to examine variations in response, but do so using only one parameter at a time and do not consider how outcomes could be impacted by the combination of all of a given patient's unique clinical characteristics [1,2]. Accordingly, the current paradigm of summarizing clinical trial results into guidelines has resulted in very slow adoption of potentially beneficial therapies and even led to a risk-treatment paradox whereby patients least likely to be treated are those with the greatest potential to benefit from said treatment [3,4].

In patients with atrial fibrillation at increased risk for stroke (as determined by $CHADS_2$-$VA_2Sc$ [5] score), current guidelines recommend anticoagulation to reduce the risk of ischemic events [6], but this decision must be weighed against the increased risk of bleeding. As many risk factors for stroke overlap with those for bleeding, the choice to anticoagulate can be complicated. Moreover, the introduction of direct oral anticoagulants (DOACs) has further increased the complexity of these decisions [7–9]. The Randomized Evaluation of Long-Term Anticoagulation Therapy (RE-LY) trial compared warfarin with dabigatran in patients with atrial fibrillation at increased risk for stroke, and found lower risk of stroke or systemic embolism and a comparable risk of major bleeding with dabigatran 150mg twice daily; however, the risk of gastrointestinal bleeding was significantly higher with dabigatran 150mg [10]. We used data from the RE-LY trial to create models predicting individual patients' risk for stroke or systemic embolism and for major bleeding with alternative doses of dabigatran as compared with warfarin, with the hope these models would provide a means for translating the results of RE-LY to clinical practice [2].

## Methods

### Patient population

RE-LY was an international randomized trial that included 18,113 patients with atrial fibrillation at 951 centers in 44 countries who were at increased risk for stroke (see S1 Appendix for

definitions). The study design, inclusion and exclusion criteria, and primary results were reported previously [10,11]. Eligible patients had documented atrial fibrillation within 6 months of enrollment, and at least one of the following characteristics that put them at increased risk for stroke: prior stroke or transient ischemic attack, a left ventricular ejection fraction (EF) of less than 40%, New York Heart Association (NYHA) class II or higher heart failure symptom within 6 months before screening, age of at least 75 years, or age of 65 to 74 years plus diabetes mellitus, hypertension, or coronary artery disease. Major exclusion criteria included severe heart-valve disorder, stroke within 14 days, severe stroke within 6 months, an increased risk of hemorrhage (see S2 Appendix), creatinine clearance (CrCl) less than 30ml per minute, active liver disease, and pregnancy. The original trial protocol was approved by review boards at all the participating sites, and all patients provided written informed consent. The 73 patients who received zero doses of study drug were excluded, leaving 18,040 patients in the analysis. Because the RE-LY database includes only de-identified patient data, this study was deemed exempt by the Yale School of Medicine and St. Luke's Mid-America Heart Institute Institutional Review Boards.

Patients were randomized to either one of two fixed doses of dabigatran (110mg twice daily or 150mg twice daily), administered in a blinded manner, or to open-label use of warfarin, titrated to an international normalized ratio (INR) of 2.0–3.0. Patients were followed for a median of 2.0 years, with 99.9% achieving complete follow up.

## Outcomes

The primary outcomes in both the original trial and our analysis were stroke or systemic embolism and major bleeding (see S3 Appendix for definitions). Secondary outcomes included stroke, systemic embolism, death, myocardial infarction, pulmonary embolism, transient ischemic attack (TIA), and hospitalization. All primary and secondary outcomes were adjudicated independently by two investigators who were blinded to treatment assignments [10].

## Statistical analyses

Baseline characteristics are reported as frequencies for categorical variables, and mean (standard deviation) and median with interquartile range (IQR) for continuous variables. The chi-square test was used to compare categorical variables, while ANOVA was used to compare continuous variables.

## Model construction

Using the RE-LY dataset, we developed separate risk models to estimate patients' risks for stroke or systemic embolism and for major bleeding. To construct each model, we identified candidate variables *a priori* for inclusion, based on the published literature and clinical experience. Only information available prior to randomization was included. We then fit multivariable logistic regression models to predict the occurrence of stroke or systemic embolism and major bleeding. To improve the feasibility of implementing the models, we sought to reduce the model to those variables that accounted for 95% of the predicted variability of the full model. To do so, we ranked all variables by their contribution to the predictive capacity of the model, then removed them sequentially starting with the variables with the least contribution to the model's variance until all variables retained in the model had at least a 95% contribution to the model's predictive capacity [12]. The initial variables considered an the variables included in the final models are listed in the S4 and S5 Appendices. We used restricted cubic splines to test the assumption that the continuous variables were linearly associated with the outcomes, and then calculated discrimination (c-statistic) and calibration using plots of

observed vs. predicted means by decile for each model [13]. No significant nonlinearity was detected.

Both the stroke/systemic embolism and major bleeding models were internally validated using bootstrap resampling for 500 replications. For each step of resampling, the model was refit as described above, and performance (calibration slope and c-statistics) was assessed on the bootstrapped data and "validated" on the original dataset. The difference in performance between the two datasets was calculated and averaged over the 500 replications. The average differences estimates due to overfitting were then subtracted from the final reported performance of the full and reduced models [14,15].

### Describing the heterogeneity of treatment effect

To describe the heterogeneity of benefits and risks of each treatment, we applied the risk models to every patient in the RE-LY trial three times: first assuming treatment with warfarin, a second time assuming treatment with dabigatran 110mg, and a third time assuming treatment with dabigatran 150mg. We calculated the absolute differences in risk between warfarin and dabigatran 110mg, and between warfarin and dabigatran 150mg for each patient. To describe the heterogeneity of treatment effect at the individual level, we calculated the absolute differences in risk between warfarin and dabigatran 110mg, and between warfarin and dabigatran 150mg for each patient. The absolute differences in the risk of stroke or systemic embolism were calculated as each patient's risk when treated with dabigatran 110mg minus the risk if treated with warfarin (for the same patient), and as the risk when treated with dabigatran 150mg minus the risk if treated with warfarin. Next, to show the heterogeneity of treatment effect at the individual level, we calculated the standard deviation (SD) of these absolute risk differences. This process was repeated to describe the absolute differences in the risk of major hemorrhage, as well as the mean and standard deviation of these absolute differences. We report a density plot to demonstrate the distribution of absolute differences between dabigatran 110mg and warfarin, and between dabigatran 150mg and warfarin across the RE-LY population. Model calibration was assessed by plotting observed event rates against deciles of predicted risk rates. We also developed integer risk scores to help illustrate the relative risk of stroke/systemic embolism and major bleeding by treatment group and clinical variables.

We strove to adhere to the standards set forth for transparency in reporting of risk models and heterogeneity analyses set forth in the TRIPOD and PATH statements [16,17]. All analyses were conducted using R version 3.3.1 (R Project for Statistical Computing, Vienna, Austria) and SAS version 9.3 (SAS Institute, Cary, NC).

## Results

The analysis included a total of 18,040 patients, of whom 5,983 were assigned to dabigatran 110mg, 6,059 were assigned to dabigatran 150mg, and 5,998 were assigned to warfarin. Baseline characteristics are shown in S1 Table and were similar across the three treatment groups. Across the entire cohort, median age of participants was 71.4 years, 63.6% were male, and 70.0% were white. Past medical history of participants included hypertension (78.8%), systolic heart failure (32.0%), diabetes mellitus (23.3%), and prior stroke/systemic embolism/TIA (21.8%).

Over the median follow-up of 2.0 years, the rates of the primary outcome of stroke/systemic embolism in the analytic cohort were 1.4%, 0.9%, and 2.1% for dabigatran 110mg, dabigatran 150mg, and warfarin, respectively. The rates of the primary outcome of major bleeding were 3.6%, 4.3%, and 4.7%, respectively (S2 Table).

## Predicting primary outcome events in RE-LY

The prediction model for stroke/systemic embolism is displayed in Fig 1, while Fig 2 shows the model for major bleeding. Both models showed acceptable discrimination, with a c-statistic of 0.68 for the stroke/systemic embolism model and 0.69 for the major bleeding model, and good calibration. For the stroke/systemic embolism model calibration, p-value for intercept is 0.8743, for slope is 0.8311; for the major bleeding model calibration, p-value for intercept is 0.9718, for slope is 0.7318. The final models included 9 variables for the stroke/systemic embolism model and 14 variables for the major bleeding model). S1 and S2 Figs show the prediction model calibration curves.

## Predicted benefits, risks, and heterogeneity of treatment effect

The mean absolute reduction in predicted risk for stroke or systemic embolism with dabigatran 110mg as compared to warfarin was 0.78% ± 0.95%. However, there was significant heterogeneity across the study population, and the range of absolute risk reduction for stroke/systemic embolism with dabigatran 110mg compared with warfarin was a decrease in risk of 9.33% to an increase in risk of 2.15%. For major bleeding, the mean absolute risk reduction with dabigatran 110mg was 1.12% ± 1.44% and ranged from a 7.77% decrease in risk to a 23.2% increase in risk with dabigatran 110mg compared with warfarin.

When comparing dabigatran 150mg to warfarin for the outcome of stroke or systemic embolism, the mean predicted absolute risk reduction was 1.32% ± 1.31% and ranged from a decrease of 11.6% to an increase of 3.30% with dabigatran 150mg compared to warfarin. For major bleeding, the mean absolute risk reduction with dabigatran 150mg was 0.41% ± 2.39% and ranged from an 8.93% decrease in risk to a 63.4% increase in risk with dabigatran 150mg compared with warfarin.

When considering patients' risks for both stroke or systemic embolism and major bleeding, there was significant variability across the study population. Fig 3A (dabigatran 110mg and warfarin) and Fig 3B (dabigatran 150mg and warfarin) show scatter plots to depict each patient's absolute difference in risk (risk with dabigatran minus risk with warfarin) of the two primary outcomes, with stroke or systemic embolism on the y-axis and major bleeding on the x-axis. Patients in the left lower quadrant of each plot experienced lower risk of both stroke/

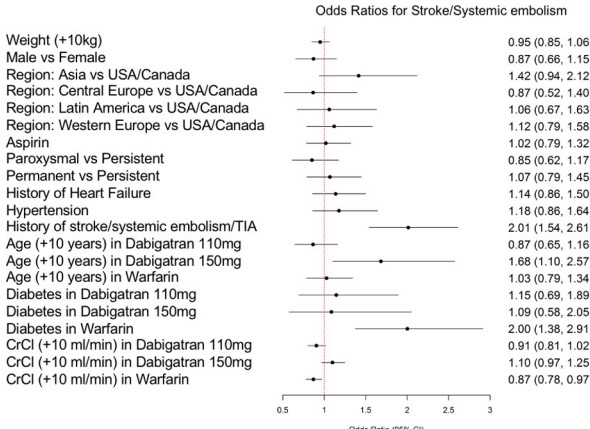

**Fig 1. Risk prediction model for stroke or systemic embolism.** Shown are the odds ratio (OR) and 95% confidence interval (CI) for each variable. For variables with interactions with treatment assignment, ORs are presented separately for treatment with dabigatran 110mg, dabigatran 150mg, and warfarin. CI, confidence interval; CrCl, creatinine clearance; TIA, transient ischemic attack; USA, United States of America.

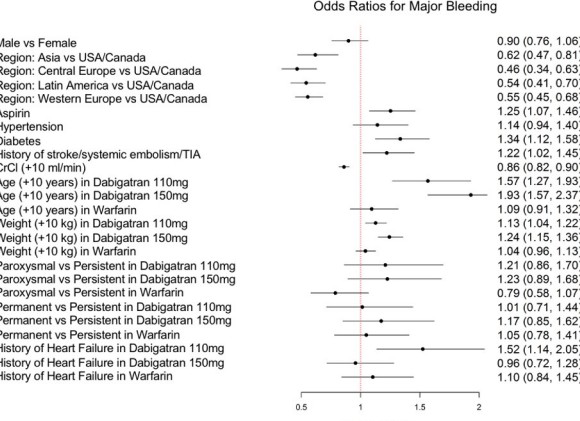

**Fig 2. Risk prediction model for major bleeding.** Shown are the odds ratio (OR) and 95% confidence interval (CI) for each variable. For variables with interactions with treatment assignment, ORs are presented separately for treatment with dabigatran 110mg, dabigatran 150mg, and warfarin. CI, confidence interval; CrCl, creatinine clearance; TIA, transient ischemic attack; USA, United States of America.

systemic embolism and bleeding risk with dabigatran, while patients in the right upper quadrant experienced lower risks of both outcomes with warfarin. The right lower quadrant represents patients with a decreased risk of stroke or systemic embolism and an increased risk of bleeding with dabigatran, and the left upper quadrant shows patients for whom dabigatran resulted in a higher risk of stroke or systemic embolism and a lower risk of bleeding.

When comparing dabigatran to warfarin, 76.5% (110mg dose) and 67.1% (150mg dose) of patients had lower risk of both stroke/systemic embolism and major bleeding; 0.4% (dabigatran 110mg) and 3.8% (dabigatran 150mg) had higher risk of both outcomes. The right lower quadrant (patients with decreased stroke/systemic embolism but higher bleeding) included 13.3% (dabigatran 110mg) and 26.3% (dabigatran 150mg) of patients.

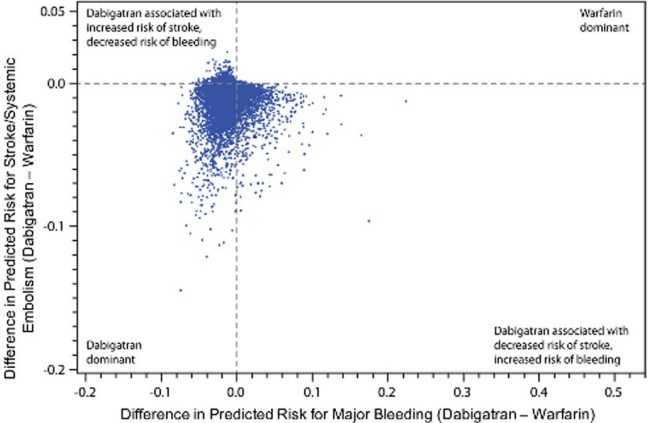

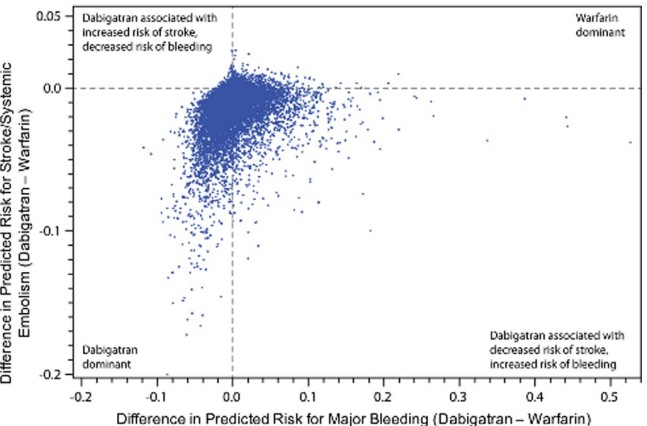

**Fig 3. Scatter plot depicting each patient's net predicted risk for stroke or systemic embolism and major bleeding. Panel A** shows the plot comparing treatment with dabigatran 110mg versus warfarin, and **Panel B** compares dabigatran 150mg and warfarin. Each dot represents a patient, with the net predicted risk (warfarin minus dabigatran) of stroke or systemic embolism on the y-axis, and the net predicted risk (warfarin minus dabigatran) of major bleeding on the x-axis. The further left on the x-axis, the lower the patient's risk of major bleeding with dabigatran, and the further down on the y-axis, the lower the patient's risk of stroke or systemic embolism.

To demonstrate the potential utility of the models in clinical care, we created sample patients and calculated their predicted risks of the primary outcomes with visual decision aids using Icon Array (Fig 4) [18]. The first patient, Patient A, is a 68-year-old, 50-kg woman from the United States who has a history of heart failure, a creatinine clearance of 49 ml/min, has paroxysmal atrial fibrillation, is not taking aspirin, and has no history of diabetes, stroke, systemic embolism, or TIA. Helen's estimated risk of stroke/systemic embolism is lower with dabigatran 150mg than with warfarin (0.43% per year vs. 1.74% per year), and her risk of major bleeding is also lower with dabigatran 150mg (4.45% per year vs. 5.26% per year with warfarin; Fig 4A). The second patient, Patient B, is a 62-year old, 85-kg man, from the United States, with a history of diabetes and TIA, a creatinine clearance of 85 ml/min, who has permanent atrial fibrillation, is taking aspirin, and has no history of heart failure. Frank's risk of stroke is also estimated to be lower with dabigatran 150mg (0.92% per year) than with warfarin (3.84% per year), however, his risk of major bleeding is higher with dabigatran 150mg (13.1% per year, Fig 4B) than with warfarin (8.43% per year). Additionally, integer risk scores for each of stroke/systemic embolism and major bleeding are shown in S3 and S4 Tables.

## Discussion

Practicing clinical medicine in a way that applies the results from robust clinical trials into everyday care is a difficult task. In this study, we developed two multivariable risk models using data from the RE-LY randomized clinical trial to predict patients' risk of stroke/systemic embolism and risk of major bleeding if treated with warfarin, dabigatran 110mg, and dabigatran 150mg, respectively. We found significant heterogeneity of treatment effects among patients in the benefits and risks of the three different treatments, with factors including age, sex, diabetes, hypertension, body weight, renal function, region of residence, and type of atrial fibrillation (among others) playing roles that shape individual patients' risks. Developing strategies, such as the risk models developed here, to convey these risks and benefits to individual patients offers the potential to optimally target new treatments to those who most benefit and to explain to patients why one treatment versus another is being recommended. For example, our analysis showed that there are patients for whom dabigatran lowers the risk of both stroke/systemic embolism and of major bleeding, others in whom there are similar benefits and risks, and yet other patients for whom dabigatran would result in increased risks of both outcomes. Simply applying the RE-LY results to all patients based on the mean treatment effect (which showed lower risk of stroke/systemic embolism and similar risk of bleeding with dabigatran 150mg compared to warfarin) would be providing sub-optimal treatment for many patients, potentially placing some at unnecessary risk and, for others, underestimating the potential benefits of dabigatran. We propose that our models, and ones like it to be produced in the future, present a significant opportunity to use modern data analysis techniques to optimize patient outcomes. This is in keeping with the Precision Medicine Initiative aims, which seek to advance medicine to a place where treatments are tailored to the patient using individualized approaches [19].

The decision to anticoagulate in the setting of atrial fibrillation is complex and has grown more so as the number of medication options has increased with the approval of the DOACs, starting with dabigatran in 2010 following publication of the RE-LY results. The evidence supporting the reduction of stroke risk with anticoagulation in select patients with atrial fibrillation is strong [20,21]. But this potential benefit must be weighed against the risk of bleeding, which can range from day-to-day bruising to major intracranial hemorrhage and life-threatening gastrointestinal bleeding. Recent work modeling individual risk in patients from the RE-LY trial, comparing the two doses of dabigatran to no anticoagulation, has shown significant heterogeneity among the trial population, with some clearly benefiting from each dose of

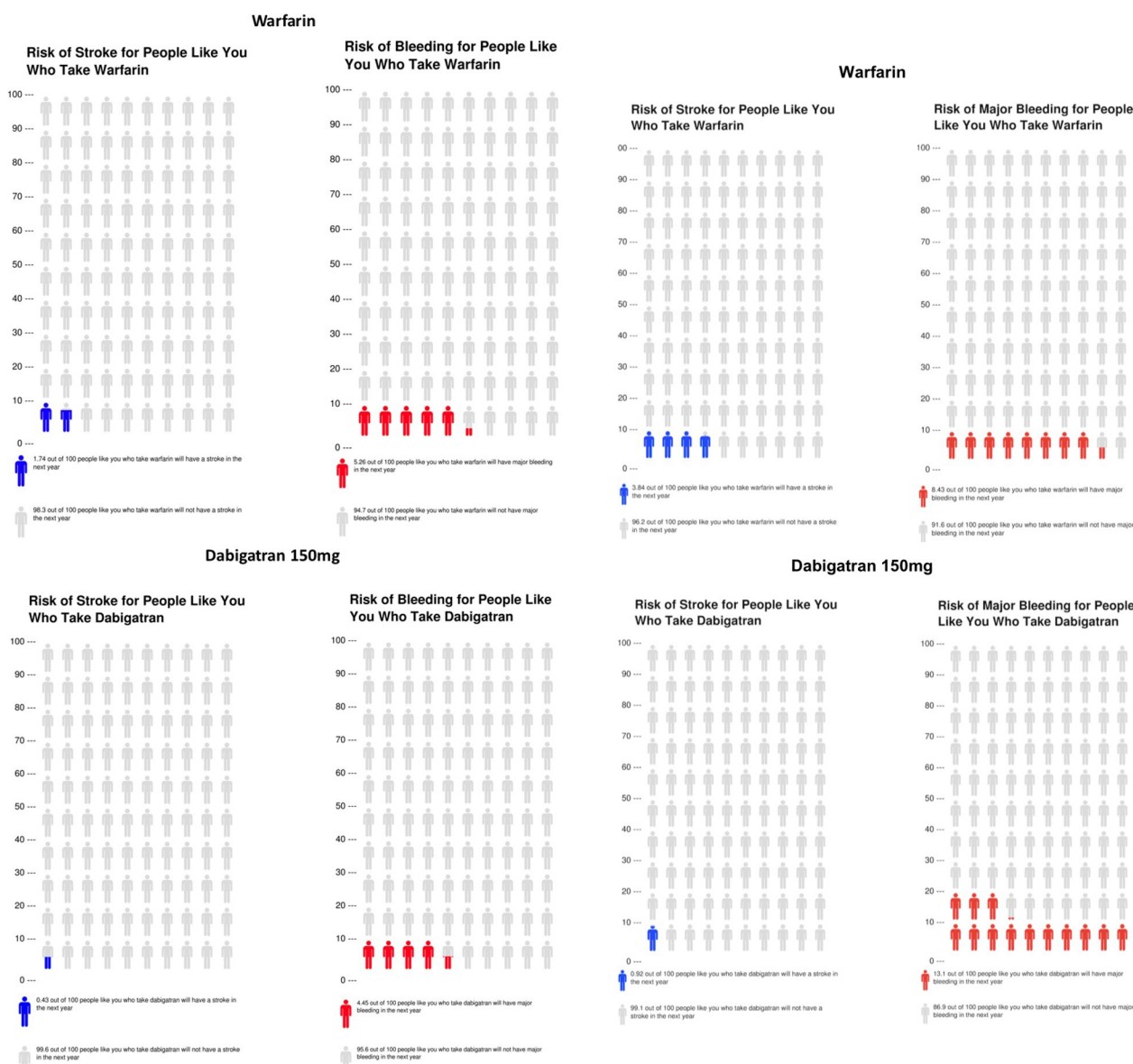

**Fig 4. Icon Array decision-aids for two model patients showing the risks of stroke or systemic embolism and major bleeding with warfarin (top) and dabigatran 150mg (bottom). Panel A** shows the risks for Patient A, a 68-year-old, 50-kg woman from the United States who has a history of heart failure, has a creatinine clearance of 49 ml/min, has paroxysmal AF, is not taking aspirin, and has no history of diabetes, stroke, systemic embolism, TIA. **Panel B** shows the risks for Patient B, a 62-year-old, 85-kg man, from the United States, with diabetes, a creatinine clearance of 85 ml/min, has a history of TIA, has permanent AF, is taking aspirin, and has no history of heart failure.

dabigatran over the other [22]. Our findings validate the individualized approach of the prior paper, but the prior study's models used variables selected by the investigators, whereas our study included all available variables in the models and removed them by backward selection [22]. For the case of stroke/systemic embolism risk, our method developed a more parsimonious model with only 9 variables included, as opposed to 14 variables in the prior study; we feel this likely improves the feasibility of implementation of our model. The prior study also focused on comparing dabigatran to no anticoagulation, while we were primarily interested in

comparisons to warfarin, as in this high-risk population it is reasonable to assume a high proportion would be anticoagulated.

By producing these multivariable models, we propose a system whereby such models could be used during clinical encounters to produce a more individualized risk/benefit profile for treatment options for each patient. The current atrial fibrillation guidelines recommend that most patients with atrial fibrillation be treated with a DOAC instead of warfarin, but this is based on the overall effects seen in the clinical trials [6]. One can imagine a scenario where, once models have been appropriately validated, a risk calculator could be built into the electronic medical record and automatically pull patient data to quickly display an individual patient risk score for clinicians to communicate to patients. Similar secondary analyses of landmark clinical trials have been performed to guide decision making for choice of antiplatelet therapy, and for intensity of blood pressure control [23,24]. Application of individualized bleeding risk models to patients undergoing percutaneous coronary intervention (PCI) have also been shown to improve patient outcomes, reducing bleeding events by 38% in adjusted analysis [25].

Relatedly, advances in decision aids, including the Icon Arrays that we used in this study, can be used to quickly and easily produce helpful visual tools for patients to better understand their unique risk profile. Such decision aids could also be automatically generated and displayed in the exam room or sent to a patient's smart phone. This type of shared decision making is in keeping with current guidelines [6], but is difficult in the current environment without readily available tools for engaging and teaching patients. Prior work in has shown that when patients are given personalized risk information regarding choice of stent type (drug-eluting vs. bare metal) prior to PCI, they have significant increases in patient knowledge and engagement in shared decision-making [26,27]. Providing patients with personalized information in the setting of anticoagulation may have a similar benefit of increasing patient engagement, which may extend to improving adherence when a treatment plan is ultimately selected [28].

## Limitations

Our study should only be interpreted in light of several potential limitations. First, our findings have not been validated in an independent sample outside of the RE-LY population. Although we performed bootstrap validations, this applies only to those meeting the inclusion and exclusion criteria of RE-LY. While modeling the heterogeneity of treatment effects is an important means for translating the results of clinical trials to individual patients, the population upon which the models are built includes only those patients meeting the inclusion and exclusion criteria of the trial. The models are thus only valid in such patients and clinicians need to use their clinical judgment as to whether or not a given patient would have been considered eligible and, if not, whether the cause for ineligibility would make the model invalid. In RE-LY, patients with higher bleeding risk were systematically excluded and caution should be used in applying the model to such patients. Of course, this also applies to the application of the original clinical trial results, which are merely summaries of mean treatment effect, and not a limitation of the models, per se. Of note, when examining the list of high-risk bleeding conditions (S2 Appendix), they include conditions that would make a clinician hesitate to start anticoagulation of any type, and thus we feel that our model applies to the large majority of patients for whom anticoagulation is being considered. Second, although our models detected evidence of heterogeneity in treatment effect, the potential exists for other interactions that went undetected in our analysis but could still affect patient outcomes. Third, the fact our population was followed in an RCT means they were likely more adherent to medications than in real-world practice; relatedly, we do not have a measure of compliance during the RE-LY trial which may

be relevant given the potential easier compliance with fixed-dose dabigatran vs. warfarin which is dosed to reach target INR values. Fourth, our models had adequate but modest c-statistics (0.675 for the stroke/systemic embolism model and 0.694 for the major bleeding model). However, it has been shown that models with c-statistics >0.60 can provide benefit to patients [29], and we feel that using our models when selecting treatment options is superior to treating all patients as if they would realize the average benefit for the study population or attempting to individualize treatment using subgroup analyses. Fifth, our models were developed using data comparing dabigatran and warfarin, and thus cannot be extrapolated to comparisons of warfarin with other DOACs; however, we would suggest that similar models be developed using data from clinical trials of other DOACs so those trial results can be better applied to individual patients in clinical practice. Finally, the models in this study only predict individualized risk for the endpoints of stroke or systemic embolism and major bleeding. Patients may also be interested in other individualized information, such as their risk of stroke alone (without systemic embolism) or of myocardial infarction with each treatment option. However, stroke or systemic embolism and major bleeding were selected for this study, as they were for RE-LY, because they represent the most significant outcomes of concern related to anticoagulant choice in patients with atrial fibrillation at an increased risk of stroke.

## Conclusions

In conclusion, using data from the RE-LY trial, we generated two multivariable models to predict individualized risk profiles for patients with atrial fibrillation treated with dabigatran or warfarin for prevention of stroke or systemic embolism. Our analysis showed significant heterogeneity of treatment effect in the RE-LY trial population, building on evidence that it exists in other trials. Although not yet validated externally, these models show promise to help personalize patient treatment decisions, and to enhance the shared decision-making process by providing patients with easy-to-understand individualized information about their predicted risks and benefits with a given treatment.

## Supporting information

**S1 Fig. Calibration curve for model predicting stroke or systemic embolism.**
(DOCX)

**S2 Fig. Calibration curve for model predicting major bleeding.**
(DOCX)

**S1 Table. Baseline characteristics of RE-LY patients by treatment group.**
(DOCX)

**S2 Table. Primary outcomes of stroke or systemic embolism and major bleeding by treatment group.**
(DOCX)

**S3 Table. Integer score for stroke or systemic embolism risk.**
(DOCX)

**S4 Table. Integer score for major bleeding risk.**
(DOCX)

**S1 Appendix. Definition of increased risk of stroke.**
(DOCX)

**S2 Appendix. Definition of increased risk for hemorrhage.**
(DOCX)

**S3 Appendix. Definition of primary endpoints.**
(DOCX)

**S4 Appendix. Variables included in stroke/systemic embolism risk model.**
(DOCX)

**S5 Appendix. Variables included in major bleeding risk model.**
(DOCX)

## Author Contributions

**Conceptualization:** Samuel W. Reinhardt, Nihar R. Desai, Yuanyuan Tang, Jeremy Ader, John A. Spertus.

**Data curation:** Nihar R. Desai, Yuanyuan Tang, Philip G. Jones, John A. Spertus.

**Formal analysis:** Samuel W. Reinhardt, Nihar R. Desai, Yuanyuan Tang, Philip G. Jones, Jeremy Ader.

**Investigation:** Samuel W. Reinhardt, Nihar R. Desai, Yuanyuan Tang, Philip G. Jones, Jeremy Ader, John A. Spertus.

**Methodology:** Samuel W. Reinhardt, Nihar R. Desai, Yuanyuan Tang, Philip G. Jones, John A. Spertus.

**Resources:** John A. Spertus.

**Supervision:** Nihar R. Desai, John A. Spertus.

**Validation:** Yuanyuan Tang, Philip G. Jones, Jeremy Ader.

**Visualization:** Jeremy Ader.

**Writing – original draft:** Samuel W. Reinhardt.

**Writing – review & editing:** Samuel W. Reinhardt, Nihar R. Desai, Yuanyuan Tang, Philip G. Jones, Jeremy Ader, John A. Spertus.

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
