## [Decision Letter · Decision Letter 0]

20 Jan 2021

PONE-D-20-38636

Personalizing the Decision of Dabigatran versus Warfarin in Atrial Fibrillation: A Secondary Analysis of the Randomized Evaluation of Long-Term Anticoagulation Therapy (RE-LY) Trial.

PLOS ONE

Dear Dr. Reinhardt,

Thank you for submitting your manuscript to PLOS ONE. After careful consideration, we feel that it has merit but does not fully meet PLOS ONE’s publication criteria as it currently stands. Therefore, we invite you to submit a revised version of the manuscript that addresses the points raised during the review process.

Both reviewers have important comments regarding the analytic approach and make suggestions for additional exploration and revision that are quite relevant. Although the importance of this work is recognised, certainly also by me, I urge you to seriously consider the suggested points. 

We look forward to receiving your revised manuscript.

Kind regards,

Hugo ten Cate, MD, PhD

Academic Editor

PLOS ONE

Journal Requirements:

2. Thank you for including your ethics statement:  "The protocol was approved by review boards at all the participating sites, and all patients provided written informed consent."

Please amend your current ethics statement to include the full name of the ethics committee/institutional review board(s) that approved your specific study

"I have read the journal's policy and the authors of this manuscript have the following competing interests: Nihar Desai works under contract with the Centers for Medicare and Medicaid Services to develop and maintain performance measures used for public reporting and pay for performance programs. He reports consulting for Amgen, Boehringer Ingelheim, Cytokinetics, Relypsa, Novartis, and SCPharmaceuticals. John Spertus reports an equity interest in Health Outcomes Sciences and consulting for Amgen, Bayer, Merck, Myokardia, Novartis, United Healthcare, BCBS of Kansas City and ownership of the copyright to the SAQ, KCCQ and PAQ. All other authors report no disclosures. "

5. Please amend your list of authors on the manuscript to ensure that each author is linked to an affiliation. Authors’ affiliations should reflect the institution where the work was done (if authors moved subsequently, you can also list the new affiliation stating “current affiliation:….” as necessary).

6. Please include a copy of Table 2 which you refer to in your text on page 8.

7. We noticed you have some minor occurrence of overlapping text with the following previous publication(s), which needs to be addressed:

http://circoutcomes.ahajournals.org/content/circcvoq/10/4/e003624.full.pdf?download=true

In your revision ensure you cite all your sources (including your own works), and quote or rephrase any duplicated text outside the methods section. Further consideration is dependent on these concerns being addressed.

Reviewers' comments:

Reviewer's Responses to Questions

**Comments to the Author**

1. Is the manuscript technically sound, and do the data support the conclusions?

Reviewer #1: Yes

Reviewer #2: Yes

2. Has the statistical analysis been performed appropriately and rigorously? 

Reviewer #1: Yes

Reviewer #2: Yes

3. Have the authors made all data underlying the findings in their manuscript fully available?

Reviewer #1: Yes

Reviewer #2: Yes

4. Is the manuscript presented in an intelligible fashion and written in standard English?

Reviewer #1: Yes

Reviewer #2: Yes

5. Review Comments to the Author

Reviewer #1: This is an interesting paper on using the data from a large trial of anticoagulation therapy to predict risks of strokes, bleeds

etc. The abstract mentions a separate model for each treatment - which (fortunately) is not what has been done here. The wording used made my heart sink, but the statistical approach here is absolutely correct. Perhaps the wording can be slightly tightened up.

One assumes the use of logistic regression here as opposed to Cox regression of time to event is justifiable by the relatively low number of events and the relatively short follow-up. However, follow-up is not apparently uniform - we need to see the range of follow-up to understand whether each patient included is really at equal risk from equal temporal exposure. If the range is not tight around the median of 2 years, then an analysis that allows for the varying length of follow-up (i.e Cox regression or equivalent) needs to be used.

Figure 1 could usefully be improved by giving the risk index for each treatment (based on the log OR) - this would naturally lead into the useful figure 4 illustration.

Reviewer #2: The authors developed 2 separate multivariable models to predict thrombotic and bleeding complications in patients with atrial fibrillation, based on the RE-LY trial data. The models show limited discriminative value, but are well calibrated. The variability of treatment effects of the different antithrombotic regimens is very clear depicted in scatter plots visualising the multivariable predicted effect on bleeding or thrombotic complications. The authors wish to enable shared decision making and facilitate more appropriate use of dabigatran.

An important limitation of this study is lack of external validation, as pointed out by the authors in the discussion. We have some extra questions.

General comments:

- Patients with high bleeding risk were excluded from the RE-LY trial. What does this mean for the external validity of this model? Should it only be considered for patients at increased thrombotic risk without high bleeding risk? Please address in discussion

- I really like the patient centred idea of the authors and the manuscript, especially figures 4A and 4B are excellent examples of what would be helpful to doctor and patient. However, in the manuscript no final version of the model, or risk score, or website is given where I can calculate this for my patient. Therefore, in the end, in clinical practise (Introduction, line 94 and 95), unfortunately, this paper won’t help unless I know how to use these models…

- Why would one use these models instead of the widely used CHADSVASc and HAS-BLED scores? It would be useful to compare these models this newly developed models.

- Is the model only suitable to use for dabigatran or also for the other NOACs versus warfarin? Since the benefits of NOAC vs warfarin is considered to be a class effect more than a dabigatran effect. Please address this in discussion.

Some parts of the methodology are not fully clear:

- Why was chosen for simple logistic regression and not for Cox proportional hazards which may be appropriate for this type of survival analysis?

- The model was created by backward selection until all variables retained in the model had at least a 95% contribution to the model’s predictive capacity. How was this contribution quantified?

- The prediction models are represented in figures 1 and 2. However, in line 191 the authors state that the final models included 9 and 14 variables. It is unclear what is shown in figures 1 and 2, since the number of predictors does not correspond. At least, the final models should be represented somewhere in the manuscript, and it should be clear what model is represented in what figure.

- The patients where modelled with assumed treatment of every treatment arm. How was this possible? Was the randomization arm forced into the models? This is not described.

6. PLOS authors have the option to publish the peer review history of their article (what does this mean?). If published, this will include your full peer review and any attached files.

Reviewer #1: No

Reviewer #2: **Yes: **Willem L. Bor

---

## [Author Response · Author response to Decision Letter 0]

27 Jun 2021

PLOS ONE

Dear Editorial Board,

 Thank you for your interest in our manuscript and the insightful comments by the reviewers. We are excited to incorporate the comments, and work towards improving the content and readability of our article. The revised manuscript reflects the input of the reviewers with revisions highlighted. Our response to the individual comments follows:

Academic Editor:

o Response: We thank the editor for the helpful links to the style requirements pages. We have corrected the style of the manuscript as well as the format and names of the files to meet PLOS ONE’s requirements. 

• Please amend your current ethics statement to include the full name of the ethics committee/institutional review board(s) that approved your specific study

o Response: We thank the editor for this comment. Our methods section references that the original RE-LY trial was approved by the institutional review board at all the trial’s participating sites. Our specific study was deemed exempt by the Yale institutional review board given it was a secondary analysis of de-identified data and not considered to be human subjects research. We have added a sentence to the methods to reflect this exemption.

• Thank you for stating the following in the Competing Interests section: "I have read the journal's policy and the authors of this manuscript have the following competing interests: Nihar Desai works under contract with the Centers for Medicare and Medicaid Services to develop and maintain performance measures used for public reporting and pay for performance programs. He reports consulting for Amgen, Boehringer Ingelheim, Cytokinetics, Relypsa, Novartis, and SCPharmaceuticals. John Spertus reports consulting for Amgen, Bayer, Merck, Myokardia, Novartis, United Healthcare, BCBS of Kansas City and ownership of the copyright to the SAQ, KCCQ and PAQ. All other authors report no disclosures. " Please confirm that this does not alter your adherence to all PLOS ONE policies on sharing data and materials, by including the following statement: "This does not alter our adherence to PLOS ONE policies on sharing data and materials.” 

o Response: We thank the editor for this comment and have added this sentence to the Competing Interests section. 

o Response: This was done as stated above.

• We note that you have indicated that data from this study are available upon request. PLOS only allows data to be available upon request if there are legal or ethical restrictions on sharing data publicly. For information on unacceptable data access restrictions, please see http://journals.plos.org/plosone/s/data-availability#loc-unacceptable-data-access-restrictions. In your revised cover letter, please address the following prompts: 

o a) If there are ethical or legal restrictions on sharing a de-identified data set, please explain them in detail (e.g., data contain potentially identifying or sensitive patient information) and who has imposed them (e.g., an ethics committee). Please also provide contact information for a data access committee, ethics committee, or other institutional body to which data requests may be sent. 

o b) If there are no restrictions, please upload the minimal anonymized data set necessary to replicate your study findings as either Supporting Information files or to a stable, public repository and provide us with the relevant URLs, DOIs, or accession numbers. Please see http://www.bmj.com/content/340/bmj.c181.long for guidelines on how to de-identify and prepare clinical data for publication. For a list of acceptable repositories, please see http://journals.plos.org/plosone/s/data-availability#loc-recommended-repositories.

o We will update your Data Availability statement on your behalf to reflect the information you provide.

Response: We thank the editor for the comment. The data are available through application to Boehringer-Ingelheim, and we as the authors do not have jurisdiction to share the data ourselves. Requests for data access can be made through Vivli (https://vivli.org/).

• Please amend your list of authors on the manuscript to ensure that each author is linked to an affiliation. Authors’ affiliations should reflect the institution where the work was done (if authors moved subsequently, you can also list the new affiliation stating “current affiliation:….” as necessary).

o Response: We thank the editor for this suggestion and have updated the author affiliations as requested.

• Please include a copy of Table 2 which you refer to in your text on page 8.

o Response: We thank the editor for this comment and apologize for the error. The text should have referred to “Supplemental Table 2”, which has now been renamed to S2 Table. The text has been updated accordingly. 

• We noticed you have some minor occurrence of overlapping text with the following previous publication(s), which needs to be addressed. In your revision ensure you cite all your sources (including your own works), and quote or rephrase any duplicated text outside the methods section. Further consideration is dependent on these concerns being addressed.

o Response: We thank the editor for this observation and apologize for the oversight. The occurrences of minor overlapping text were unintentional and have been corrected.

Reviewer #1:

• The abstract mentions a separate model for each treatment - which (fortunately) is not what has been done here. The wording used made my heart sink, but the statistical approach here is absolutely correct. Perhaps the wording can be slightly tightened up.

o Response: We thank the reviewer for this helpful suggestion. The text of the abstract has been updated to read the following: “we developed multivariable models to predict the risk for stroke or systemic embolism and for major bleeding including all three treatment groups (dabigatran 110mg, dabigatran 150mg, and warfarin)”

• One assumes the use of logistic regression here as opposed to Cox regression of time to event is justifiable by the relatively low number of events and the relatively short follow-up. However, follow-up is not apparently uniform - we need to see the range of follow-up to understand whether each patient included is really at equal risk from equal temporal exposure. If the range is not tight around the median of 2 years, then an analysis that allows for the varying length of follow-up (i.e Cox regression or equivalent) needs to be used.

o Response: We thank the reviewer for the comment and agree that this is an important consideration. We agree with the reviewer that imbalances in follow-up could invalidate the results of logistic regression. Unfortunately, we no longer have access to the data to assess this concern. However, given the rigorous procedures under which this clinical trial was conducted, we find it unlikely that there would be systematic biases in follow-up duration.

• Figure 1 could usefully be improved by giving the risk index for each treatment (based on the log OR) - this would naturally lead into the useful figure 4 illustration.

o Response: We thank the reviewer for this suggestion. We have added an integer score to the Supplemental Appendix (S3 and S4 Tables) to help model risk for each treatment, taking into account clinical variables.

Reviewer #2

• Patients with high bleeding risk were excluded from the RE-LY trial. What does this mean for the external validity of this model? Should it only be considered for patients at increased thrombotic risk without high bleeding risk? Please address in discussion

o Response: The reviewer raises an important issue, which is that the model is built upon the patients enrolled in the study. Accordingly, we have added the following to the Discussion: 

“While modeling the heterogeneity of treatment effects is an important means for translating the results of clinical trials to individual patients, the population upon which the models are built includes only those patients meeting the inclusion and exclusion criteria of the trial. The models are thus only valid in such patients and clinicians need to use their clinical judgment as to whether or not a given patient would have been considered eligible and, if not, whether the cause for ineligibility would make the model invalid. In RE-LY, patients with higher bleeding risk were systematically excluded and caution should be used in applying the model to such patients. Of course, this also applies to the application of the original clinical trial results, which are merely summaries of mean treatment effect, and not a limitation of the models, per se. Of note, when examining the list of high-risk bleeding conditions (S2 Appendix), they include conditions that would make a clinician hesitate to start anticoagulation of any type, and thus we feel that our model applies to the large majority of patients for whom anticoagulation is being considered.” 

• I really like the patient centred idea of the authors and the manuscript, especially figures 4A and 4B are excellent examples of what would be helpful to doctor and patient. However, in the manuscript no final version of the model, or risk score, or website is given where I can calculate this for my patient. Therefore, in the end, in clinical practise (Introduction, line 94 and 95), unfortunately, this paper won’t help unless I know how to use these models…

o Response: We agree with the reviewer that this is an important question to address. We have added the integer score risk models to the supplemental appendix (S3 and S4 Tables) to increase their accessibility.

• Why would one use these models instead of the widely used CHADSVASc and HAS-BLED scores? It would be useful to compare these models this newly developed models.

o Response: We thank the reviewer for the comment. The important innovation of our approach is to explicitly estimate the outcomes as a function of treatment. The CHADSVASc and HAS-BLED scores do not suggest how patients with different risks of thromboembolic events or bleeding might be harmed or helped with the use of dabigatran or warfarin. This is explicitly estimated by our approach and can be used to engage patients in shared decision-making about which anti-thrombotic regimen to choose. 

• Is the model only suitable to use for dabigatran or also for the other NOACs versus warfarin? Since the benefits of NOAC vs warfarin is considered to be a class effect more than a dabigatran effect. Please address this in discussion.

o Response: We thank the reviewer for this question. Given the heterogeneity of outcomes in clinical trials of NOACs, we would hesitate to extrapolate our models to the use of NOACs other than dabigatran. We have added sentences to the limitations to reflect this important clarification, but would also suggest that clinical trials of other NOACs be used to develop similar models so their results can be better applied to individual patients in clinical practice.

• Why was chosen for simple logistic regression and not for Cox proportional hazards which may be appropriate for this type of survival analysis?

o Response: We thank the reviewer for the comment and agree that this is an important consideration. We chose logistic regression in order to avoid potentially complex issues regarding the proportional hazards assumption in the Cox model particularly in the presence of multiple interacting factors with treatment.

• The model was created by backward selection until all variables retained in the model had at least a 95% contribution to the model’s predictive capacity. How was this contribution quantified?

o Response: The purpose of this approach was to reduce the number of variables required – a critically important consideration in improving the feasibility of prospective use of our models in clinical care – without sacrificing the predictive power of the models. We applied the approach recommended by Frank Harrell in Regression Modeling Strategies. Secaucus, NJ: Springer-Verlag New York, Inc; 2006. To develop more parsimonious models while preserving optimal discrimination, we began with full models that included all prespecified predictor variables and interactions. All variables were then ranked by their contribution to the predictive capacity of the model, and variables explaining the least variance in the model were sequentially removed until removing another variable would reduce the predictive capacity of the reduced model by >5% relative to the full model. With this approach, the reduced model accounts for >95% of the prediction capacity of the full model. 

• The prediction models are represented in figures 1 and 2. However, in line 191 the authors state that the final models included 9 and 14 variables. It is unclear what is shown in figures 1 and 2, since the number of predictors does not correspond. At least, the final models should be represented somewhere in the manuscript, and it should be clear what model is represented in what figure.

o Response: We thank the reviewer for this comment and agree that this issue needed to be clarified. The forest plots in Figures 1 and 2 depict the variables and odds ratios (with p-values) for our “pseudo-full” models, which include all the variables tested minus non-significant interaction terms. The final, parsimonious models included the 9 and 14 variables as referenced in the methods and results, after variables from the pseudo-full models with small interaction terms have been eliminated, but still allow the model to retain 95% of its predictive accuracy. However, displaying a forest plot for the final model would not be statistically sound, since after variables have been eliminated, the odds ratios and p-values no longer have statistical validity.

• The patients where modelled with assumed treatment of every treatment arm. How was this possible? Was the randomization arm forced into the models? This is not described.

o Response: We thank the reviewer for this question. Once we completed the models, we then applied the model to each patient in the trial assuming that they were treated with warfarin and we ran the model again assuming that they were treated with dabigatran 110mg. This was then repeated assuming that the patients were treated with 150mg of dabigatran. This allowed us to compare the treatment differences for each individual patient included in the trial.

In conclusion, thank you for the opportunity to revise the manuscript. We look forward to hearing from you.

Corresponding Author

Samuel W Reinhardt, MD

Section of Cardiovascular Medicine

Yale School of Medicine

789 Howard Ave, 3rd Floor

New Haven, CT, 06510

Phone: 203-873-7807

Fax: 530-316-5947

Email: samuel.reinhardt@yale.edu

---

## [Decision Letter · Decision Letter 1]

5 Aug 2021

Personalizing the decision of dabigatran versus warfarin in atrial fibrillation: A secondary analysis of the Randomized Evaluation of Long-term anticoagulation therapY (RE-LY) trial.

PONE-D-20-38636R1

Dear Dr. Reinhardt,

We’re pleased to inform you that your manuscript has been judged scientifically suitable for publication and will be formally accepted for publication once it meets all outstanding technical requirements.

Kind regards,

Hugo ten Cate, MD, PhD

Academic Editor

PLOS ONE

Additional Editor Comments (optional):

Although one reviewer did not give further feedback to your rebuttal, the statistical reviewer is satisfied and I also think that your answers and further changes are satisfactory.

Reviewers' comments:

Reviewer's Responses to Questions

**Comments to the Author**

1. If the authors have adequately addressed your comments raised in a previous round of review and you feel that this manuscript is now acceptable for publication, you may indicate that here to bypass the “Comments to the Author” section, enter your conflict of interest statement in the “Confidential to Editor” section, and submit your "Accept" recommendation.

Reviewer #1: All comments have been addressed

2. Is the manuscript technically sound, and do the data support the conclusions?

Reviewer #1: (No Response)

3. Has the statistical analysis been performed appropriately and rigorously? 

Reviewer #1: (No Response)

4. Have the authors made all data underlying the findings in their manuscript fully available?

Reviewer #1: (No Response)

5. Is the manuscript presented in an intelligible fashion and written in standard English?

Reviewer #1: (No Response)

6. Review Comments to the Author

Reviewer #1: (No Response)

7. PLOS authors have the option to publish the peer review history of their article (what does this mean?). If published, this will include your full peer review and any attached files.

Reviewer #1: No

---

## [Editor Report · Acceptance letter]

11 Aug 2021

PONE-D-20-38636R1 

Personalizing the decision of dabigatran versus warfarin in atrial fibrillation: A secondary analysis of the Randomized Evaluation of Long-term anticoagulation therapY (RE-LY) trial. 

Dear Dr. Reinhardt:

I'm pleased to inform you that your manuscript has been deemed suitable for publication in PLOS ONE. Congratulations! Your manuscript is now with our production department. 

Kind regards, 

on behalf of

Professor Hugo ten Cate 

Academic Editor

PLOS ONE